# Pathologic Response of Associated Ductal Carcinoma In Situ to Neoadjuvant Systemic Therapy: A Systematic Review

**DOI:** 10.3390/cancers15010013

**Published:** 2022-12-20

**Authors:** Umar Wazir, Neill Patani, Nahed Balalaa, Kefah Mokbel

**Affiliations:** 1The London Breast Institute, Princess Grace Hospital, London W1U 5NY, UK; 2Sheikh Shakhbout Medical City (SSMC) & Mayo Clinic, Abu Dhabi P.O. Box 11001, United Arab Emirates

**Keywords:** neoadjuvant therapy, breast cancer, ductal carcinoma in-situ, down-staging

## Abstract

**Simple Summary:**

Traditionally, the presence of ductal carcinoma in situ (DCIS) with invasive breast cancer meant that the patient may require complete removal of the breast in order to completely remove the disease. Recently there has been some evidence to the contrary. In this article, we have reviewed the current published literature to determine the rate at which DCIS was eradicated by chemotherapy and endocrine therapies administered prior to surgery, which we determined to be 40.5% and 15% respectively. This suggests that in patient who respond well to pre-surgical systemic therapy, breast conserving surgery (BCS) could be offered. This should lessen patient anxiety and morbidity.

**Abstract:**

Contrary to traditional assumptions, recent evidence suggests that neoadjuvant systemic therapy (NST) given for invasive breast cancer may eradicate co-existent ductal carcinoma in-situ (DCIS), which may facilitate de-escalation of breast resections. The aim of this systematic review was to assess the eradication rate of DCIS by NST given for invasive breast cancer. Searches were performed in MEDLINE using appropriate search terms. Six studies (N = 659) in which pathological data were available regarding the presence of DCIS prior to neoadjuvant chemotherapy (NACT) were identified. Only one study investigating the impact of neoadjuvant endocrine therapy (NET) met the search criteria. After pooled analysis, post-NACT pathology showed no residual DCIS in 40.5% of patients (267/659; 95% CI: 36.8–44.3). There was no significant difference in DCIS eradication rate between triple negative breast cancer (TNBC) and HER2-positive disease (45% vs. 46% respectively). NET achieved eradication of DCIS in 15% of patients (9/59). Importantly, residual widespread micro-calcifications after NST did not necessarily indicate residual disease. In view of the results of the pooled analysis, the presence of extensive DCIS prior to NST should not mandate mastectomy and de-escalation to breast conserving surgery (BCS) should be considered in patients identified by contrast enhanced magnetic resonance imaging (CE-MRI).

## 1. Introduction

One of the enduring challenges in the surgical management of invasive breast cancer (IBC) is the adverse impact of co-existing extra-tumoral ductal carcinoma in-situ (DCIS) on decision making. In particular, the indication for therapeutic mastectomy is frequently driven by the extent and distribution of associated non-invasive disease. The proportion of tumours with an in situ component adjacent to IBC ranges from 33% [1] to 53% [2]. Resection volumes are consequently increased to achieve satisfactory clearance margins. Furthermore, the need for margin re-excision is approximately 25% for IBC alone, but can be increased to one third of cases with associated DCIS [3].

Another pertinent issue related to co-existent DCIS with invasive ductal carcinoma (IDC) [IDC-DCIS] is the interpretation of residual DCIS with regard to the response status of patients after neoadjuvant systemic treatment (NST). This issue affects a significant subset of patients. In a retrospective survey of 20,495 cases, Ploumen et al. reported the persistence of DCIS in 4.3% of patients after neoadjuvant therapy for IBC (ypTis) [4]. The implications of ypTis compared to no residual disease (ypT0) in the interpretation of pathological complete response (pCR) has been the subject of much inquiry, given the implications for further treatment planning and prognosis. Residual DCIS in patients who achieve complete eradication of invasive disease in the breast and axillary nodes does not adversely affect local recurrence or survival. Hence, inclusion of patients with residual DCIS in the definition of pCR is justified when this outcome is used as an early surrogate for long-term survival [5].

Since DCIS is a non-invasive disease where cells possess physiological mechanisms affording protection from cytotoxic drugs in a manner similar to normal cells, it has been historically assumed that this precludes response to systemic chemotherapy or targeted therapy [6]. However, recent evidence suggests that DCIS can respond to NST and the downstaging of non-invasive disease can facilitate breast conserving surgery (BCS) after NST in patients with co-existent extensive DCIS. In this article, the literature regarding the response of IDC-DCIS to NST, including both neoadjuvant chemotherapy (NACT) and neoadjuvant endocrine therapy (NET), is systematically reviewed.

## 2. Materials and Methods

For the purposes of this systematic review, MEDLINE and The United States National Library of Medicine (NLM) database of the biomedical literature were searched using the PubMed engine and the following terms: ‘DCIS and chemotherapy’, ‘DCIS and endocrine therapy’, ‘DCIS trastuzumab’, ‘DCIS pertuzumab’, ‘DCIS neoadjuvant chemotherapy’, ‘DCIS neoadjuvant systemic therapy’, and ‘DCIS neoadjuvant endocrine therapy’.

The eligibility criteria of the systematic review were as follows: (1) Patients diagnosed with breast cancer undergoing neoadjuvant systemic therapy including chemotherapy, anti-HER2 therapy, checkpoint inhibitors or endocrine therapy, (2) Pathological data regarding the presence of DCIS before and after NST were available, (3) Prospective, retrospective and randomised controlled trials (RCTs), (4) Language restrictions in English, and (5) Sample size of the study was more than 50 cases.

Exclusion criteria included the following: (1) Abstracts, case reports and lectures, (2) Preoperative systemic therapy prescribed for short duration (less than 4 weeks) to assess change of biomarkers, (3) Incorrect or incomplete data that were unable to be extracted from other relevant studies, (4) Duplicate publications. 

Articles that met the criteria were retrieved for full-text evaluation and the relevant patient level data extracted and compiled for down-stream analysis. Subsequent pooled and subgroup analyses were performed to reflect the principal clinical phenotypes of disease.

### Statistical Analysis

Chi-square test was used to compare the eradication rate between HER2 positive triple negative breast cancer subgroups, and between NET and NACT. Yate’s correction was also used to correct the error introduced by assuming that the discrete probabilities of frequencies in the table can be approximated by a continuous distribution.

A *p*-value of 0.05 was accepted as the threshold of statistical significance.

## 3. Results

### 3.1. Literature Search

The literature search identified 1412 articles pertaining to NACT, of which six studies satisfied the aforementioned inclusion and exclusion criteria. These studies encompassed 659 patients in which pathological data regarding the presence of DCIS before and after NACT were available, detailed in Table 1 [1,7,8,9,10,11].

### 3.2. Effect of NACT

Pooled analysis of the data showed that DCIS was eradicated by NACT in 267 of 659 patients, yielding a rate of 40.5% (95% CI: 36.8–44.3). In these cases, the final post-NST histology confirmed no residual DCIS, where pre-NST biopsy showed adjacent or intra-tumoral DCIS. Notably, the rate of eradication was highest (50.8%) in those with HER2-positive disease, particularly when the DCIS was also HER2-positive [9]. There were three studies which focused on HER2-positive breast cancer (Sun et al., Groen et al., von Minckwitz et al.) [7,9,10]. Among this subset, DCIS was eradicated in 140 of 306 patients (45.8%, 95% CI: 40.2–51.3). In the small subgroup of TNBC, DCIS was eradicated in 35 of 78 cases (45%) [11].

### 3.3. Effect of NET

There were 332 articles identified regarding NET, of which only one [12] satisfied the inclusion and exclusion criteria. In this study, 9/59 patients (15.3%, 95% CI: 6.1–24.4) achieved complete eradication of DCIS after NET for pure DCIS [12]. Although significantly lower than that observed for HER2-positive disease (Chi-square statistic with Yates correction 8.6, *p* = 0.003), NET could nonetheless achieve eradication of DCIS in some cases.

### 3.4. Subgroup Analyses

Subgroup analyses were performed reflecting the principal clinical phenotypes of disease receiving NST.

#### 3.4.1. NACT vs. NET 

The DCIS eradication rate was significantly greater in HER2-positive disease (Chi-square statistic with Yates correction 17.8, *p* = 0.00003) and TNBC treated with NACT (Chi-square statistic with Yates correction 12.2, *p* = 0.0005), compared to ER-positive DCIS treated with NET. This suggests that NET is less effective in eradicating DCIS than regimens containing chemotherapy with or without targeted anti HER2 antibodies.

#### 3.4.2. HER2 Positive vs. TNBC 

Interestingly, there was no statistically significant difference identified when directly comparing HER2-positive and TNBC (46% and 45% respectively; Chi-square statistic with Yates correction 0.0001, *p* = 0.99). Although the study may have lacked sufficient power to identify small differences, large variations between these subsets would be expected to show.

## 4. Discussion

This systematic review of the literature, followed by pooled and subset analyses, clearly demonstrates that contrary to historical assumptions, DCIS can respond favourably to NST and be completely eradicated in a significant proportion of cases. Underpinning the frequency and extent of response appears to be the tumour biology of DCIS. The susceptibility of non-invasive disease to NST is predicted to some extent by the immuno-phenotype, approximating the intrinsic molecular subtype. Overexpression of HER2 was associated with a higher incidence of co-existing DCIS [13] and a higher rate of eradication [9]. Subtype analyses confirmed that the eradication rate of DCIS was significantly higher in HER2-positive disease treated by NACT than ER-positive/HER2-negative DCIS treated by NET (46% vs. 15%, *p* = 0.003). This observation may be attributed to the biology of the tumour or more likely to the use of targeted biological therapy consisting of monoclonal antibodies against HER2 rather than chemotherapy. Although there was limited data regarding the effect of NST on DCIS associated with TNBC, the rate of eradication observed was comparable to HER2-positive disease (45% vs. 46%, *p* = 0.99), again significantly higher than ER-positive disease. The recent advent of checkpoint inhibitors, increasingly used in the neoadjuvant setting for TNBC >2cm, is likely to further increase the eradication rate of co-existing DCIS [11]. This contemporary targeted biological therapy was not used in any of the studies analysed, hence the potential eradication rate may have been under-estimated.

This systematic review has several notable limitations. Firstly, the studies included were mostly retrospective and employed heterogeneous pathology protocols. Secondly some of the studies included intra-tumoral DCIS, the biology of which could differ from adjacent DCIS. Thirdly, the absence of DCIS in post-NST histology may reflect the in-situ component being entirely removed during diagnostic biopsy, rather than any impact of NST on DCIS. Fourthly, it is very likely that the presence of DCIS before commencing NST was underestimated, thus leading to underestimation of the response rate [8]. Fifthly, our study did not specifically examine the relationship between DCIS grade and proliferation index and response rate to NST. However, as per invasive disease, it is plausible that high-grade DCIS with a high Ki67 index is more likely to respond than lower grade disease with a lower Ki67 [11]. Sixthly, although we investigated the impact of NST on DCIS eradication rate, the potentially greater rate of partial response/down-staging which could impact upon surgical decision making was not examined. Seventhly, this study does not provide any data regarding long-term clinical outcomes and assumes that histological eradication is sufficient to justify the oncological safety of de-escalating breast cancer surgery. Finally, not all studies provided detailed information regarding the molecular profile of the DCIS in relation to ER and HER2 status and the proportion of HER2-positive disease (46%) was higher in the pooled analysis than encountered in clinical practice. Furthermore, the methodology of evaluating the pathological extent of DCIS prior to, and after NST was not standardised across all studies.

The presence of extensive DCIS in association with invasive breast cancer is often considered to be an absolute indication for mastectomy, even for patients receiving NST. This dogma is based on the widely held assumption that DCIS does not respond well to NST. However, the data presented in this study demonstrate that 40% of DCIS can be eradicated by NST. There is now compelling evidence to challenge this paradigm and breast conserving surgery (BCS) should be the preferred option for excellent responders. This selected group of patients can be identified by post-NST contrast-enhanced MRI, demonstrating the impact of NST on previously enhancing non-invasive disease. The accumulating evidence that BCS with radiotherapy is associated with better overall survival than mastectomy alone, further strengthens the argument for conservative surgery wherever possible [14,15]. In addition to the complete eradication of IDC-DCIS, achieving partial response/down-staging of extensive DCIS can also significantly impact upon surgical decision-making (Figure 1, Figure 2, Figure 3 and Figure 4) [16]. HER2-positivity, absence of suspicious calcifications on mammography, treatment with dual HER2-blockade, near/complete response on MRI, absence of calcifications in DCIS on pre-NST biopsy and Ki67 expression > 20% in DCIS, were all factors predicting response to NST in univariate analysis [7].

The literature search identified only one study pertaining to DCIS eradication in response to NET. A single arm phase II clinical trial investigated the impact of neoadjuvant letrozole on pure ER-positive DCIS. After six months of treatment, the authors observed an eradication rate of 15%, significantly lower than that observed for NST that includes chemotherapy and targeted therapy for invasive breast cancer associated with DCIS [12]. Extensive DCIS is often associated with widespread mammographic micro-calcifications that persist after NST despite histologically confirmed eradication of DCIS [1]. In such cases, adequate sampling of residual micro-calcification through BCS to exclude residual viable DCIS, should be considered an adequate alternative to mastectomy, as illustrated in Figure 2 and Figure 4. Alternatively, the residual micro-calcifications can be subjected to post-NST image-guided vacuum biopsy/excision prior to surgery to ensure DCIS eradication. In addition to facilitating BCS, eradication or down-staging of DCIS may permit nipple preservation in women undergoing skin-sparing mastectomy where disease extends close to the nipple before NST (Figure 4).

There is a growing body of evidence that residual DCIS alone (i.e., ypTis) does not adversely impact overall survival [5,17]. Osdoit et al. recently reported data from the I-SPY2 response-adaptive randomised trial. It was observed that among patients who achieved pCR, there was no significant difference in event-free survival (EFS), distant recurrence-free survival (DRFS), or local recurrence rate (LRR) based on presence or absence of residual DCIS [17]. This is concordant with the findings reported by Mazouni et al. in a retrospective study involving 2302 patients who underwent NST between 1980 to 2004 [5]. This is particularly relevant to the era of treatment de-escalation where omission of surgery is being considered in exceptional responders to NST [18,19]. This underscores the need to develop a robust system to assess the presence of residual DCIS following NST. Contrast-enhanced MRI and post-NST image-guided vacuum biopsy are being currently evaluated in this context [20]. Among imaging modalities, breast MRI can potentially distinguish between ypT0 and ypTis after NCT, especially in patients with TNBC [21].

Recent studies have demonstrated that in triple negative DCIS, tumour infiltrating lymphocytes express high levels of PD-L1 (>50% cells), indicating potential susceptibility to PD-L1 antagonists [22]. In patients with early-stage TNBC, neoadjuvant treatment with atezolizumab in combination with nab-paclitaxel and anthracycline-based chemotherapy significantly improved pCR rates with acceptable safety [23]. However, the study did not examine the impact of this PD-L1 antagonist on the eradication rate of DCIS, which warrants detailed analysis. Moreover, the potential impact of other targeted biological therapies such as poly-adenosine diphosphate–ribose polymerase (PARP) inhibitors on DCIS associated with invasive breast cancer should be investigated in suitable patients [24]. Finally, future studies should also investigate the potential efficacy of the next generation antibody-drug conjugate trastuzumab deruxtecan (Enhertu) on DCIS components. This drug has been shown to be effective in targeting both high and HER2-low positive breast cancer in the metastatic setting [25].

The potential role of liquid biopsy using circulating tumour DNA or circulating tumour cells platforms should be investigated in future studies as a tool of assessment of completeness of resection [26].

## 5. Conclusions

Based on this systematic review of the literature and subtype-specific analysis, it is reasonable for individual tumour boards to consider de-escalation of breast surgery in carefully selected patients with extensive co-existent DCIS showing excellent response to NST especially in patients with HER2 positive and triple negative breast cancer.

We propose that such patients undergo breast CE-MRI surveillance for two years following BCS in addition to standard annual mammographic screening. Areas of abnormal enhancement on MRI or change in the microcalcifications would represent an indication for imaging guided tissue sampling 

This personalised approach, particularly in the context of HER2-positive and triple-negative breast cancer, can potentially spare patients the physical and psychological morbidity of unnecessary mastectomy.

## Figures and Tables

**Figure 1 cancers-15-00013-f001:**
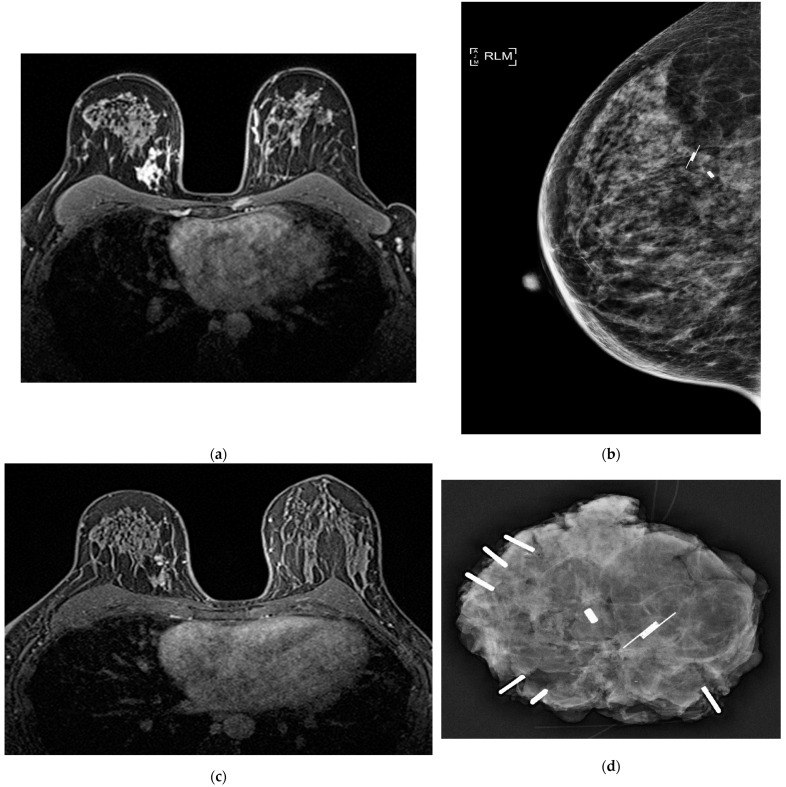
(**a**) Contrast-enhanced MRI of a 40-year-old woman with a 15 mm invasive breast cancer associated with non-mass enhancement (total footprint 60 mm) confirmed to be high grade DCIS on MRI-guided biopsy. The tumour was HER2 positive. (**b**) Pre-operative mammogram showing marker clip and SAVI SCOUT reflector. (**c**) Post-NST contrast-enhanced MRI showing excellent response of both invasive and DCIS components to paclitaxel, transtuzumab & pertuzumab, justifying de-escalation to BCS. (**d**) Intra-operative specimen mammogram showing radiological marker clip and SAVI SCOUT reflector centrally, with ligaclips for orienting circumferential margins. (**e**) Post-operative appearance following BCS and adjuvant radiotherapy. Mastectomy was avoided by the down-staging of DCIS by NST.

**Figure 2 cancers-15-00013-f002:**
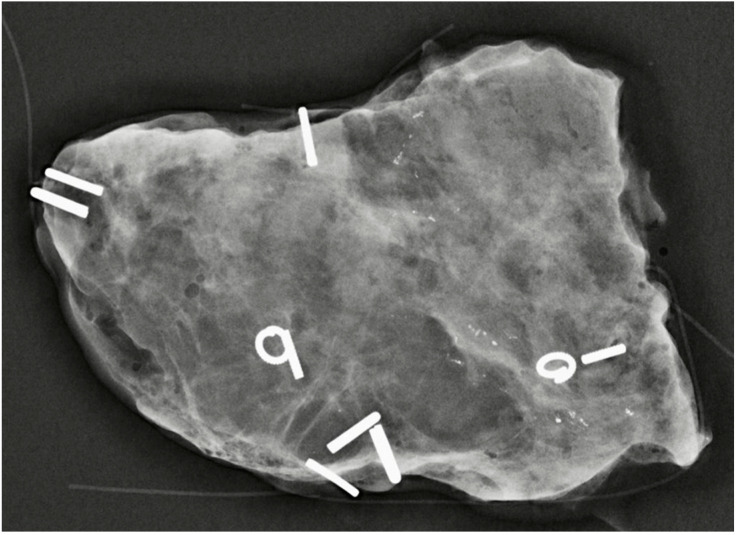
Intra-operative specimen mammogram showing residual micro-calcification after NST for triple-positive invasive breast cancer associated with DCIS with orientation staples in situ. The patient had BCS and the final histology confirmed pCR. There was no DCIS associated with the micro-calcifications seen. The patient remains disease-free three years after surgery.

**Figure 3 cancers-15-00013-f003:**
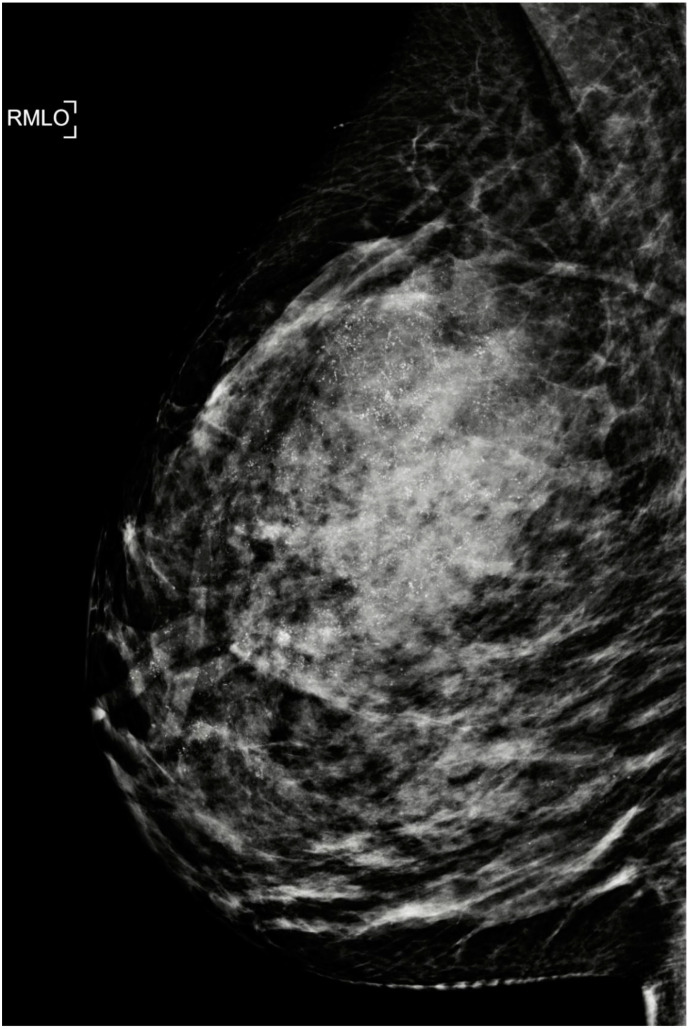
Right oblique mammogram showing extensive micro-calcifications (120 mm) associated with high grade DCIS (ER-negative/HER2-positive) and invasive breast cancer (triple-positive) in a 37-year-old patient. The final histology showed pCR and no viable residual DCIS after NST (chemotherapy and HER2 antibodies). The patient had nipple-sparing mastectomy and immediate reconstruction with targeted axillary dissection. Preservation of the nipple was facilitated by the eradication of extensive DCIS with NST.

**Figure 4 cancers-15-00013-f004:**
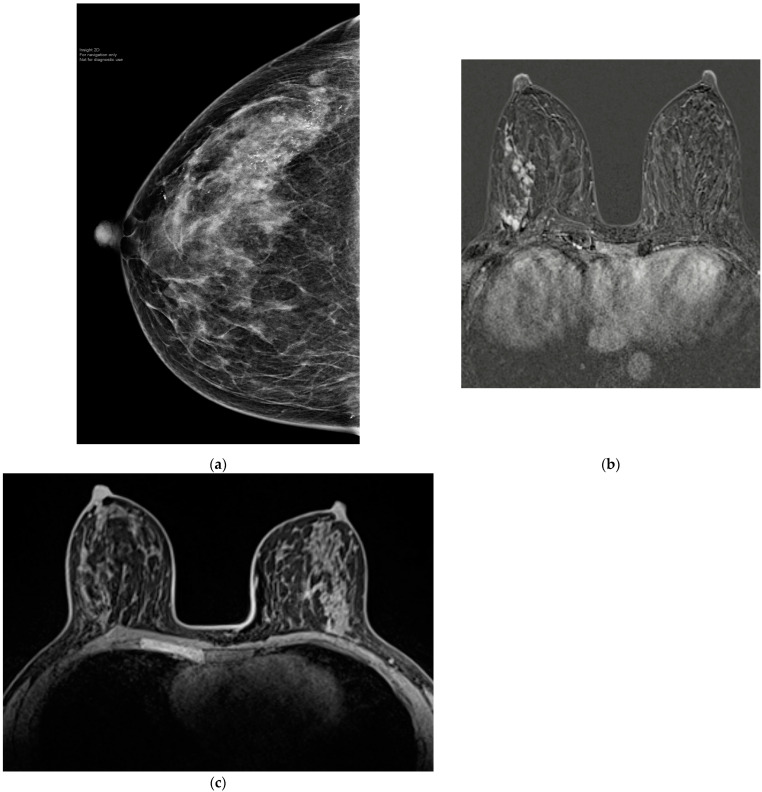
(**a**) 50-year-old patient with 15 mm triple-positive grade two invasive ductal carcinoma in the right upper outer quadrant with extensive biopsy proven DCIS and pleomorphic micro-calcifications within 1 cm of the nipple. (**b**) MRI showed extensive mass and non-mass enhancement spanning 10 cm. The patient received neoadjuvant weekly Paclitaxel and trastuzumab with pertuzumab for three months. (**c**) Post-NST MRI showed a complete radiological response. The patient underwent nipple-sparing mastectomy and reconstruction with SLNB. The final histology confirmed pCR and no viable DCIS associated with residual micro-calcifications.

**Table 1 cancers-15-00013-t001:** Six studies regarding residual DCIS after NST, satisfying the inclusion and exclusion criteria after systematic review of the literature. In pooled analysis, 40.5% (267/659) patients achieved complete eradication of DCIS after NST. DCIS: ductal cancer in situ; IDC-DCIS: co-existent DCIS with invasive ductal carcinoma; NST: neoadjuvant systemic therapy; pCR: pathological complete response.

Citation	Author (Year)	Study Design	Patients (N)	Overall pCR	IDC-DCIS	DCIS-specific pCR
[7]	Groen (2021)	Prospective	316	46%	138	64
[8]	Labrosse (2021)	Retrospective	1148	19.4% (283)	225	82
[9]	von Minckowitz (2021)	RCT (subset)	158	50.8% (30)	59	30
[10]	Sun (2019)	Retrospective	280	36.4% (102)	129	46
[1]	Goldberg (2017)	Prospective	92	42% (39)	30	10
[11]	Van la Parra (2018)	Prospective	328	36.9%	78	35
**Total**					**659**	**267**

## Data Availability

Not applicable.

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
