# Peer review of "Pathologic Response of Associated Ductal Carcinoma In Situ to Neoadjuvant Systemic Therapy: A Systematic Review"

_cancers, 2022, doi:10.3390/cancers15010013_

Round 1
Reviewer 1 Report
This manuscript evaluates a pertinent clinical issue on whether ductal carcinoma in situ (DCIS) responds to neoadjuvant systemic therapy (NST). It is a systemic review, having performed a Medline search on articles with certain key words and synthesizing the data obtained. The findings are that there is evidence, although not in abundance, that DCIS does respond to NST. Downstaging of DCIS size can allow de-escalation of surgery and reduce the need for mastectomy for patients presenting initially with extensive DCIS. However, there is still room for further research as to whether DCIS responds to newer systemic drugs and selection criteria for breast conservation instead of mastectomy.
I am in complete agreement with the authors that no effort should be spared to reduce the use of mastectomy for breast cancer but this de-escalation must be performed safely. In line with this, I have a few questions for the authors:
1. 1 In women with widespread microcalcifications denoting the extensive distribution of DCIS, how does one propose follow up of these calcifications if breast conserving surgery (BCS) is performed to clear pathologic margins but residual microcalcifications? This question refers in particular to the patient illustrated in Figure 3, if breast conservation were to have been performed instead of a mastectomy. The natural choice in this case would be MRI but this may conflict with the desire to reduce mastectomy rates as research has shown that MRI tends to increase the likelihood of mastectomy.1
2. In line with question 1, what is the selection criteria for the use of MRI in such cases, bearing in mind, again, that MRI could result in a triage to mastectomy?
3. As a curiosity, why did the patient illustrated in Figure 1 have both a marker clip and a SaviScout reflector relatively close to each other in the same breast? I note also several marker clips in close proximity in the specimen radiograph for the patient illustrated in Figure 2 as well.
4. Are there blood, tissue and molecular markers which may act as selection criteria for adequate resection of DCIS in the presence of residual microcalcifications after BCS and clear pathologic margins?
Thank you for your work.
Reference
1. Newman LA. Role of preopearative MRI in the management of newly diagnosed breast cancer patients. J Am Coll Surg 2020;230:331-339
Author Response
1. In women with widespread microcalcifications denoting the extensive distribution of DCIS, how does one propose follow up of these calcifications if breast conserving surgery (BCS) is performed to clear pathologic margins but residual microcalcifications? This question refers in
particular to the patient illustrated in Figure 3, if breast conservation were to have been performed instead of a mastectomy. The natural choice in this case would be MRI but this may conflict with the desire to reduce mastectomy rates as research has shown that MRI tends to increase the likelihood of mastectomy.
We accept this criticism and have added this paragraph to the manuscript:
We propose that such patients undergo breast CE-MRI surveillance for two years following BCS in addition to standard annual mammographic screening. Areas of abnormal enhancement on MRI or change in the microcalcifications would
represent an indication for imaging guided tissue sampling.
2. In line with question 1, what is the selection criteria for the use of MRI in such cases, bearing in mind, again, that MRI could result in a triage to mastectomy?
This does not apply to such patients since the detection of residual or recurrent disease during follow-up will mandate mastectomy in view previous BCS and radiation therapy.
3. As a curiosity, why did the patient illustrated in Figure 1 have both a marker clip and a SaviScout reflector relatively close to each other in the same breast? I note also several marker clips in close proximity in the specimen radiograph for the patient illustrated in Figure 2 as well.
The marking clip close to the Savi Scout reflector in the centre of the specimen was inserted at the time of biopsy to mark the site of the tumour. The peripheral clips (single: superior, two clips: medial and 3: inferior) are deployed after resection in order to orientate the specimen on the radiograph.
4. Are there blood, tissue and molecular markers which may act as selection criteria for adequate resection of DCIS in the presence of residual microcalcifications after BCS and clear pathologic margins?
We accept this criticism and have added this paragraph to the manuscript:
The potential role of liquid biopsy using circulating tumour DNA or circulating
tumour cells platforms should be investigated in future studies as a tool of
assessment of completeness of resection (Ref: Crook, T.; Leonard, R.; Mokbel, K.;
Thompson, A.; Michell, M.; Page, R.; Vaid, A.; Mehrotra, R.; Ranade, A.; Limaye,
S.; Patil, D.; Akolkar, D.; Datta, V.; Fulmali, P.; Apurwa, S.; Schuster, S.;
Srinivasan, A.; Datar, R. Accurate Screening for Early-Stage Breast Cancer by
Detection and Profiling of Circulating Tumor Cells. Cancers 2022, 14, 3341.
https://doi.org/10.3390/cancers14143341).
Reviewer 2 Report
Wazir U. et al. conducted a systematic review of the current literature to assess the eradication rate of DCIS associated with invasive breast cancer following neoadjuvant chemotherapy and neoadjuvant hormone therapy. The authors conclude that neoadjuvant therapy can also eradicate the in situ component of the tumour allowing conservative breast surgery. To this end, the authors perform a selection of both retrospective and prospective studies for neoadjuvant chemotherapy (including different therapy regimens based on the heterogeneous molecular structure) and a single study of neoadjuvant hormone therapy.
The topic is of marginal importance given the current consolidated clinical practice. Furthermore, the analysis carried out is compromised by the lack of some fundamental information, such as the receptor status of DCIS, and the lack of uniformity in the method of instrumental staging pre- and post-neoadjuvant therapy in the different studies.
Some further points should be addressed:
-Tables summarising the results should be added
- Page3, lines 110-111; Why was HER2-positive disease and TNBC compared with ER-positive DCIS treated with NET (instead of NACT)?
-Line 184 please correct “micro-classification”
The English language and style require minor spell check.
Author Response
We have added the following to the section of the discussion regarding limitations:
Not all studies provided detailed information regarding the molecular profile of the DCIS in relation to ER and Her-2 status. Furthermore, the methodology of
evaluating the pathological extent of DCIS prior to, and after NST was not
standardised across all studies.
Some further points should be addressed:
-Tables summarising the results should be added.
Table 1 does this to a great extent.
-Page3, lines 110-111; Why was HER2-positive disease and TNBC compared with ER-positive DCIS treated with NET (instead of NACT)?
The comparison demonstrates that NET is less effective in eradicating DCIS than
regimens containing chemotherapy with or without targeted anti HER2 antibodies. We are adding this sentence to the section of the manuscript as well.
-Line 184 please correct “micro-classification”
Done
-The English language and style require minor spell check.
Done
Reviewer 3 Report
This study is really an interesting topic. As mentioned in the article, results from this review analysis was contrary to our historical assumptions, that DCIS can respond favorably to NST and be completely eradicated in a significant proportion of cases. If the applied statistical method is correct, in my personal opinion, different voices should be allowed to sound. But I think the study lacked sufficient power to identify the response of DCIS could impact upon surgical decision. It would be better if the author could modify the title of the article to a more appropriate one.
Author Response
We have modified to the title of the paper, the following:
Pathologic response of associated ductal carcinoma in-situ to neoadjuvant
systemic therapy: a systematic review.
Round 2
Reviewer 2 Report
Dear authors,
The topic is of marginal importance given the current consolidated clinical practice